# Rapid Scan Electron Paramagnetic Resonance using an EPR-on-a-Chip Sensor

Silvio Künstner[1], Anh Chu[2], Klaus-Peter Dinse[1,5], Alexander Schnegg[3], Joseph E. McPeak[1], Boris Naydenov[1], Jens Anders[2,4*], Klaus Lips[1,5*]

[1]Berlin Joint EPR Laboratory and EPR4Energy, Department Spins in Energy Conversion and Quantum Information Science (ASPIN), Helmholtz-Zentrum Berlin für Materialien und Energie GmbH, Hahn-Meitner-Platz 1, 14109 Berlin, Germany
[2]Institute of Smart Sensors, Universität Stuttgart, 70569 Stuttgart, Germany
[3]EPR4Energy, Max-Planck-Institut für chemische Energiekonversion, 45470 Mülheim an der Ruhr, Germany
[4]Center for Integrated Quantum Science and Technology (IQST), Stuttgart and Ulm, Germany
[5]Berlin Joint EPR Laboratory, Fachbereich Physik, Freie Universität Berlin, 14195 Berlin, Germany
*contributed equally

*Correspondence to*: Boris Naydenov (boris.naydenov@helmholtz-berlin.de)

**Abstract.** Electron paramagnetic resonance (EPR) spectroscopy is the method of choice to investigate and quantify paramagnetic species in many scientific fields, including materials science and the life sciences. Common EPR spectrometers use electromagnets and microwave (MW) resonators, limiting their application to dedicated lab environments. Here, novel aspects of voltage-controlled oscillator (VCO) based EPR-on-a-Chip (EPRoC) detectors are discussed, which have recently gained interest in the EPR community. More specifically, it is demonstrated that with a VCO-based EPRoC detector, the amplitude-sensitive mode of detection can be used to perform very fast rapid scan EPR experiments with a comparatively simple experimental setup to improve sensitivity compared to the continuous-wave regime. In place of a microwave (MW) resonator, voltage-controlled oscillator (VCO)-based EPRoC detectors use an array of injection-locked VCOs, each incorporating a miniaturized planar coil as a combined microwave source and detector. A striking advantage of the VCO-based approach is the possibility to replace the conventionally used magnetic field sweeps with frequency sweeps with very high agility and near-constant sensitivity. Here, proof-of-concept rapid scan EPR (RS-EPRoC) experiments are performed by sweeping the frequency of the EPRoC VCO array with up to 400 THz s$^{-1}$, corresponding to a field sweep rate of 14 kT/s. The resulting time-domain RS-EPRoC signals of a µm scale BDPA sample can be transformed into the corresponding absorption EPR signals with high precision. Considering currently available technology, the frequency sweep range may be extended to 320 MHz, indicating that RS-EPRoC shows great promise for future sensitivity enhancements in the rapid scan regime.

## 1 Introduction

Electron Paramagnetic Resonance (EPR) spectroscopy is a widespread analytical tool for studying species with unpaired electrons relevant in chemistry, physics, biology, and medicine. The main uses of EPR are the quantification of paramagnetic centers (Eaton et al., 2010) in, e.g., chemical analyses or quality control, the identification and characterization of radicals (Villamena, 2017), paramagnetic defects (Brodsky and Title, 1969), and transition metal ion states (Van Doorslaer and Vinck, 2007) in biological samples, semiconductors, and during chemical reactions for assignment of the electronic and atomic structure of paramagnetic states (Neese, 2017).

In conventional continuous wave (CW) EPR spectrometers, a microwave (MW) cavity resonator with a high quality factor ($Q$) is used to enhance the signal-to-noise ratio (SNR) and the resolution. The resonator couples the magnetic field component of the MW (~9.4 GHz in X-band spectrometers) to the magnetic moments of the unpaired electron spins of the sample. The response of the magnetic susceptibility of the sample is detected via the reflected MW using an MW bridge. To achieve the resonance condition, an external magnetic field, $B_0$, is swept linearly and continuously while the MW frequency is kept constant due to the very low bandwidth of the resonator, as dictated by the high $Q$ employed to increase SNR. In standard continuous wave (CW, CW-EPR) operation, the magnetic field is modulated, enabling lock-in detection. Presently, EPR

spectrometers are relatively bulky, having typical dimensions ranging from several tens of cm for smaller benchtop X-band systems to several meters for higher resolution research spectrometers. While the formers are limited to X-band operation, high-end spectrometers are available at much higher frequencies, operating at X- (9 GHz), Q- (36 GHz), W- (94 GHz) bands up to even higher frequencies (~263 GHz). Sales prices of EPR spectrometers range from ≈50 k€ for benchtop devices up to well over 1 M€ for high-end spectrometers. However, for more widespread use of this powerful technique in science, industry, and even consumer applications, access to portable, cost-effective, and easy-to-operate EPR sensors is required. In the optimum case, such a spectrometer would consist of a single sensor that can be immersed in, attached to, or embedded in a sample of interest, removing the limitations of current resonator-based techniques. This vision requires a complete redesign of the EPR spectrometer, in which the bulky electromagnets and microwave parts are replaced by smaller permanent magnets and miniaturized electronic components capable of sweeping the frequency at a fixed magnetic field. An important challenge in designing such frequency-swept EPR systems is to ensure a (near-) constant sensitivity over wide sweep ranges.

In pursuit of this redesign, EPR spectrometers have been developed that enable more flexible *operando* applications such as a hand-held EPR system for transcutaneous oximetry (Wolfson et al., 2015), an EPR "dipstick" spectrometer that can be immersed in an aqueous solution (Zgadzai et al., 2018), and the EPR Mobile Universal Surface Explorer (EPR-MOUSE) as a field-swept, surface-sensitive EPR spectrometer (Switala et al., 2017). In all of these designs, however, a conventional microwave bridge is used for MW generation and detection, limiting their applicability to dedicated laboratories. Moreover, the sensitivity as a function of operating frequency is still dictated by the characteristics of the utilized resonator.

Significant progress in semiconductor fabrication technology has propelled the design of new EPR spectrometers that are fully integrated into a single silicon microchip, so-called EPR-on-a-Chip (EPRoC) devices (Yalçin and Boero, 2008; Anders et al., 2012; Yang and Babakhani, 2015; Handwerker et al., 2016; Zhang and Niknejad, 2021). These EPRoC devices either integrate a conventional microwave bridge or variants of it in a single integrated circuit (Yang and Babakhani, 2015; Zhang and Niknejad, 2021), use a fixed-frequency oscillator (Yalçin and Boero, 2008; Anders et al., 2012) or a voltage-controlled oscillator (VCO) (Handwerker et al., 2016) to detect the EPR signal. In the latter approach, a miniaturized coil with a diameter of a few hundred micrometers is embedded into a voltage-controllable LC oscillator to serve both as microwave source and EPR detector. The idea of using a VCO instead of a microwave bridge to excite and detect the nuclear magnetic resonance (NMR) signal was first proposed in 1950 (Pound and Knight, 1950). Importantly, this approach circumvents the classical trade-off between resonator $Q$ and detection sensitivity (Hyde et al., 2010), enabling frequency-swept EPR over wide frequency ranges with near-constant sensitivity. This allows the use of permanent magnets for smaller, more affordable, battery-driven spectrometers, as recently demonstrated (Handwerker et al., 2016; Schlecker et al., 2017a, b; Anders and Lips, 2019). The magnetic field strengths of practical permanent magnets (<1.5 T) limit the EPR excitation frequency to below 35 GHz, limiting the use of very high-frequency EPRoC detectors to research applications (Matheoud et al., 2018). In addition to allowing for the design of miniaturized, battery-driven "conventional" EPR spectrometers, EPRoC detectors can also easily be integrated into complex and application-specific sample environments, opening the door to numerous potential *in situ* and/or *operando* EPR applications from room temperature to cryogenic temperatures down to 4 K (Gualco et al., 2014).

To further increase the sensitivity of the EPR technique, especially for samples with long relaxation times, the rapid scan EPR (RS-EPR) technique has been introduced (Eaton and Eaton, 2016). The advantage of the RS technique as compared to CW EPR is that much higher microwave excitation fields ($B_1$) can be applied to the sample before saturation effects are observed. The RS technique overcomes MW saturation limitations of the spin system by spending less time on resonance. Thereby, the SNR can be significantly enhanced in comparison to traditional CW-EPR (Eaton and Eaton, 2016). This is accomplished by scanning the magnetic field or MW frequency quickly such that the resonance is passed in a time shorter than the relaxation times $T_1$ and $T_2^*$. The EPR signal is recorded with a transient digitizer instead of a phase-sensitive detector, and passage effects may appear as "wiggles" on the trailing edge of the EPR resonance signals in the time domain. The passage effects can then be removed by *Fourier* deconvolution to recover the conventional slow-passage EPR spectrum (Stoner et al., 2004; Joshi et

al., 2005b; Tseitlin et al., 2011a), i.e., the sample susceptibility. There are various reports on enhanced SNR of RS-EPR compared to CW-EPR using spin-trapped radicals (Mitchell et al., 2013a), nitroxyl radicals (Mitchell et al., 2012), irradiated fused quartz (Mitchell et al., 2011a), and samples with long relaxation rates such as hydrogenated amorphous silicon (a-Si:H) (Mitchell et al., 2013b; Möser et al., 2017) where the latter showed an improvement in spin sensitivity of more than one order of magnitude. In addition, RS-EPR allows for the determination of spin relaxation times, which is particularly useful in very high-frequency EPR and under conditions where pulse EPR techniques are not applicable (Laguta et al., 2018). In most of the aforementioned experiments, field-swept RS-EPR was employed. Sweeping magnetic fields at high rates over a wide range is technically demanding and requires specialized coils and high current, high slew rate amplifiers. The realistically achievable maximum sweep width is limited to about 20 mT at slow rates (tens of kHz), restricting field-swept RS-EPR to the quite narrow spectra of the aforementioned sample classes (organic radicals, samples with low $g$ anisotropy and small hyperfine interaction, etc.). Many transition metal ion states in biological and other samples, however, have much larger spectral widths. For faster rates, the sweep width is limited even more for typical resonator sample sizes. Additionally, vibrations of the coils and eddy currents induced in the metallic parts of the resonator may distort the spectrum, which may be especially large for fast, wide sweeps (Joshi et al., 2005a). The sweep width limitation of field-swept RS-EPR can be overcome using the non-adiabatic rapid sweep (NARS) (Kittell et al., 2011) or field-stepped direct detection (FSDD) EPR technique (Yu et al., 2015). This technique, however, complicates the data acquisition as well as the post-processing, prolongs the measurement time and necessitates the use of an electromagnet. Employing frequency-swept RS-EPR circumvents these problems, however, routinely used high $Q$, low bandwidth resonators limit the achievable sweep width considerably. With EPRoC, it is possible to utilize frequency-swept RS-EPR over large sweep widths of more than 1.8 GHz (63 mT) (Chu et al., 2017) without the constraints of resonator-based RS-EPR and thus may be used for interrogation of $g$ and $A$ anisotropy of samples with large hyperfine splitting and long relaxation times, such as in transition metal complexes at cryogenic temperatures, with increased sensitivity compared to CW-EPR using a small-footprint EPRoC spectrometer with a permanent magnet. Rapid scan operation with single-chip integrated oscillators was initially proposed in Gualco et al., 2014; however, no details about detecting the resulting EPR signal were provided. The fact that the tuning voltage of a VCO can be used to produce fast frequency ramps is well known and has been previously used in RS-EPR (Laguta et al., 2018). However, VCO-based EPRoC detectors also provide a very interesting means of detecting the resulting change in sample magnetization, which was first proposed in Chu et al. 2017). In this report, we extend the approach proposed in Chu et al. (2017) for RS-EPRoC experiments to allow for a reproducible reconstruction of the slow-passage spectrum from the RS data. Embedding the VCO into a high-bandwidth phase-locked loop (PLL) allows for a precise definition of the phase of the exciting $B_1$-field from an external reference, even in the presence of temperature and other experimental fluctuations. Moreover, the amplitude-sensitive mode of detection with an implicit, high-bandwidth AM demodulator built directly into the LC VCO, as suggested in Chu et al. (2017), is used to detect the sample magnetization with a high bandwidth on the order of a few hundreds of MHz. Together with the very recent results from Chu et al. (2021), a closed theory for the analysis of the AM RS-EPRoC signals is provided.

Experimentally, proof-of-concept frequency-swept RS-EPR experiments (Tseitlin et al., 2011b; Hyde et al., 2010) with a sweep width of 128 MHz (4.57 mT) using an RS-EPRoC detector are reported and an improvement of almost two orders of magnitude in SNR was observed compared to CW-EPRoC measurements conducted with the same detector.

## 2 Materials and methods

### 2.1 EPR-on-a-Chip setup

The schematic of the employed experimental setup is depicted in Fig. 1. The EPRoC detector is located on a printed circuit board (PCB) which is inserted between the poles of an electromagnet (*Bruker* B-E 25) (Fig. 1a). The electromagnet was used solely because of immediate availability, without using the sweeping capabilities, and, in principle, a permanent magnet can

be used instead. A small permanent magnet for the EPRoC is currently being developed. An EPRoC design with an array of twelve injection-locked VCOs was used (see Fig. 1b), similar to the design in Chu et al. (2018). Importantly, the injection locking of $N$ VCOs lowers the phase noise of the joint array frequency by $\sqrt{N}$ (Chu et al., 2018). The utilized EPRoC detector has a frequency sweep range extending from 12.0 to 14.4 GHz (sweep width 2.4 GHz or 85.6 mT). Two techniques may be

used for detecting the spin response with the EPRoC, namely amplitude-sensitive detection (AM) (Chu et al., 2017; Matheoud et al., 2018; Chu et al., 2021) and frequency-sensitive detection (FM) (Yalçin and Boero, 2008; Anders et al., 2012). The AM and FM signals correspond to the EPR-induced changes in the VCO amplitude and frequency, respectively. While the FM signal purely represents the real component of the complex susceptibility, the AM signal represents a mixture of the imaginary, $\chi''$, and real, $\chi'$, components of the magnetic susceptibility (Chu et al., 2021). More specifically, the EPR-induced frequency

changes, $\Delta\omega_{\mathrm{osc}}$, and amplitude changes, $\Delta A_{\mathrm{osc}}$, in the AM and FM detection modes can be written as:

$$\Delta A_{\mathrm{osc}} \propto Q\chi'' - \chi' \tag{1}$$

$$\Delta\omega_{\mathrm{osc}} \propto \chi', \tag{2}$$

where $Q$ is the quality factor of the LC tank inside the VCO. Note that the FM signal only depends on $\chi'$ (Eq. (2)) and that, depending on the quality factor, the AM signal is primarily observed as an absorption signal according to $\chi''$, which is slightly distorted by the dispersion signal $\chi'$. In the experiments performed in this report, the amplitude detection mode of the VCO-based detector (cf. Fig. 1d) is employed, and the EPR signal is measured as a change in the oscillation amplitude of the VCO

(Chu et al., 2017). Although both detection modes provide theoretically the same sensitivity (Anders, 2011; Matheoud et al., 2018), the practical advantage of detecting the AM signal is that a wideband AM demodulator can be easily integrated into an LC tank VCO as suggested in Chu et al., 2017, which greatly reduces the experimental complexity. The resulting change in amplitude of oscillation of the VCO, $\delta A(t)$, is given by, cf. (Chu et al., 2017):

$$\delta A(t) \approx -\frac{Q_{\mathrm{coil}}}{\alpha_{\mathrm{od}}-1} \cdot \sin(\omega_{\mathrm{osc}}t) \cdot \frac{d}{dt}\int_{V_s} \mathbf{B_u} \cdot \mathbf{M_s}\mathrm{dV}, \tag{3}$$

where $Q_{\mathrm{coil}}$ is the unloaded factor of the LC resonator inside the VCO, $\alpha_{\mathrm{od}}$ is a design parameter ranging for practical VCOs

between two and five, $\omega_{\mathrm{osc}}$ is the VCO oscillation frequency, $\mathbf{B_u}$ is the unitary magnetic field of the VCO tank inductor, and $\mathbf{M_s}$ is the sample magnetization. Here, it should be noted that, assuming that $\omega_{\mathrm{osc}} \approx \omega_{\mathrm{L}}$, i.e. that the oscillation frequency is close to the Larmor frequency of the electron spin ensemble, Eq. (3) contains a low frequency component that corresponds to the spin magnetization in the rotating frame of reference $\mathbf{M_{s,rot}}$, and a component around twice the Larmor frequency. The implicit AM demodulator (denoted as $V_x$ in Fig. 1d) extracts the low frequency component of Eq. (3) with a sensitivity $S_{\mathrm{AM}}$

and an effective noise figure, which will be discussed later in the context of the experimental results. In principle, as suggested in (Matheoud et al., 2018), an external AM demodulator can be used instead.

The AM detection scheme is implemented in one VCO inside the injection-locked VCO array, which is used as the EPR detector for all EPR experiments shown in this paper, cf. Fig. 1. The MW frequency of the EPRoC array is controlled by a phase-locked loop (PLL) with a bandwidth of about 10 MHz and a radio frequency (RF) generator (*Rohde & Schwarz*

SMB100A) as the PLL frequency reference. As mentioned above, the PLL is crucial to derive the phase of the $B_1$ field produced by the VCO from a well-defined reference, even in the presence of fluctuations of the experimental conditions. On the EPRoC, a 32-divider is placed such that the reference frequency for the PLL is around 420 MHz (13.44 GHz / 32).

The $B_1$ magnitude may be varied by controlling the bias current, $I_{\mathrm{bias}}$, applied to the VCO with a minimum $B_1$ of about 27 µT resulting from the minimum bias current (~5 mA) required for stable oscillations of the VCO. All EPR measurements were

160 performed as a frequency swept experiment with the EPRoC detector at a central microwave frequency of 13.44 GHz and at an external magnetic field of $B_0 = 479.4$ mT. For CW-EPRoC detection, sinusoidal frequency modulation is applied to the MW carrier wave with a modulation rate $f_{\mathrm{m}}$ and a peak-to-peak modulation amplitude $\Delta f_{\mathrm{m,pp}} = 2\Delta f_{\mathrm{m}}$ (see Eq. (7), below). The CW-EPRoC signal is detected with a lock-in amplifier (*Anfatec eLockIn 203*) and is linearly baseline-corrected using the

outermost 5% of the recorded spectrum where no signal is present. For RS-EPRoC measurements, a complex transient signal
was constructed from the AM signal by invoking the *Kramers-Kronig[1]* relationship to allow accurate deconvolution and
reconstruction of the EPR spectrum (Tseitlin et al., 2010). Only the AM signal was considered due to the large demodulation
bandwidth of the implicit AM demodulator. This greatly facilitates AM RS-EPR experiments using EPRoC detectors compared
to FM RS-EPRoC, where a much larger PLL bandwidth (~80 MHz) would be needed to demodulate the FM RS-EPR and
make it available at the VCO tuning voltage. Such large PLL bandwidths are hard to be achieved due to the very high required
reference frequencies. (See Appendix B for more information concerning the bandwidth calculation).

A single grain of a α,γ-Bisdiphenylene-β-phenylallyl (BDPA, 1:1 with benzene from Sigma Aldrich, ~1.6 µg, ~2 x $10^{15}$ spins)
was placed in the AM1 coil of the EPRoC detector (see Fig. 1b). The sample volume was calculated to be 6.7 x $10^{-4}$ mm$^3$
(0.67 nL) (for more information, see Appendix F). BDPA gives an EPR signal at g = 2.003 with a linewidth of about 0.07 mT
(Meyer et al., 2014).

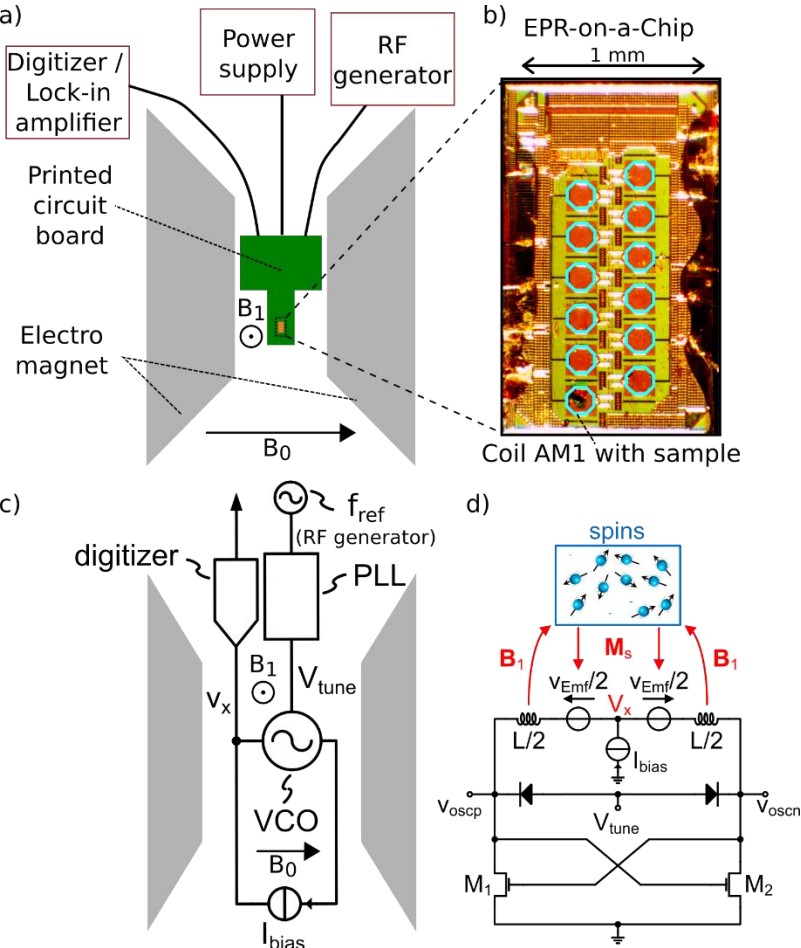

**Figure 1: a) Depiction of the EPRoC setup. The EPRoC is located on the PCB which is inserted between the poles of the electromagnet. It is connected to a signal generator, a power supply and either a lock-in amplifier (LIA) for CW measurements or digitizer for RS operation. The directions of the static $B_0$ field and $B_1$ MW field are indicated by the arrows. b) Close-up of the EPRoC array with the twelve octagonal coils. The BDPA sample is placed in the coil AM1, where an AM signal can be detected. c) A block diagram of the EPRoC setup as shown in a). The RF generator provides a reference frequency $f_{ref}$ to the phase-locked loop in which the VCO of the EPRoC is embedded. The VCO is biased using a bias current, $I_{bias}$. The AM signal at $V_x$ is detected by the digitizer. d) Illustration of the interaction between VCO-based detector and the spins in the sample. The knot $V_x$ provides the implicit AM demodulation as described in the text. The transistors $M_1$ and $M_2$ are a cross-coupled pair that acts as a "negative" resistance replenishing the energy loss of the LC tank (upper part of the electrical circit). The copyright statement only applies to d) © [2017] IEEE. Reprinted, with permission, from Chu et al. (2017).**

---

[1] Corresponding to a Hilbert transform of the signal.

## 2.2 Rapid scan using EPRoC

In RS-EPRoC operation, sinusoidal frequency modulation is applied to the fixed MW frequency, similar to CW-EPRoC operation; however, in the case of RS-EPRoC, much larger modulation rates, $f_m$, and frequency deviations, $\Delta f_m$, are used with the transient response detected directly and without lock-in amplification. The RS-EPRoC signal is recorded using a transient digitizer (*Zurich Instruments* UHF-LIA) with a sampling rate set to 450 MHz. For the baseline correction of the transient RS signal, a non-resonant transient RS background signal was recorded at a magnetic field of 400 mT and was subsequently subtracted from the experimental transient RS-EPRoC signal.

To ensure operation in the rapid passage regime as defined by Weger (1960), the scan rate $\alpha_{rot}$ of the MW frequency $\omega_{mw} = 2\pi f_{mw}$ must fulfill the following condition,

$$\alpha_{rot} = \frac{d\omega_{mw}}{dt} \gg \frac{|\gamma|B_1}{\sqrt{T_1 T_2}} \tag{4}$$

where $\gamma$ is the gyromagnetic ratio of the spin, and $B_1$ is the amplitude of the MW excitation field. The criterion for a frequency sweep to reach the non-adiabatic rapid passage regime only depends on $B_1$ according to

$$\frac{d\omega_{mw}}{dt} \gg \gamma^2 B_1^2. \tag{5}$$

as defined by Powles (1958). For sinusoidal frequency sweeps, which are used in all RS-EPRoC experiments reported in this report, the excess[2] instantaneous microwave frequency, $f_i$, is defined as

$$f_i = \Delta f_m \cos(2\pi f_m \tau), \tag{6}$$

where $\Delta f_m$ is the modulation amplitude in Hz and $f_m$ is the modulation frequency in Hz. In one scan period $T$, resonance is achieved twice, namely at $\tau = T/4$ and at $\tau = 3T/4$ where the scan rate, $\alpha$, reaches a maximum

$$\alpha = \frac{\alpha_{rot}}{2\pi} = \left.\frac{df_i}{dt}\right|_{max} = 2\pi f_m \Delta f_m \tag{7}$$

The maximum modulation amplitude in these experiments was limited by the RF generator, which provides a frequency-modulated reference signal at 420 MHz to the EPRoC via the PLL, corresponding to 13.44 GHz on the chip due to the 32-divider as mentioned above. At this frequency, the maximum frequency modulation amplitude of the RF generator (also referred to as frequency deviation) is 2 MHz, corresponding to $\Delta f_m = 32 \cdot 2$ MHz $= 64$ MHz (2.28 mT) at the VCO output frequency, which was used in the experiments reported. The maximum modulation frequency of the RF signal generator is 1 MHz; thus only about 5% of the available frequency sweep range of the EPRoC, about 2.4 GHz ($\Delta f_m \approx 1.2$ GHz, sweep width 85.6 mT), was used. This in turn limited the maximum scan rate, $\alpha$, to 402.1 THz s$^{-1}$, corresponding to 14.4 kT/s.

## 3 Results and discussion

### 3.1 Comparison between CW- and RS-EPRoC spectra

An example of a full cycle transient AM RS-EPRoC signal recorded with a bias current of 7 mA ($B_1 \sim 45.5$ µT) and a scan rate of 80 THz s$^{-1}$ is depicted in Fig. 2a where the characteristic "wiggles" resulting from the non-adiabatic rapid passage are clearly observed. Since the resonance is passed twice in each full cycle, the signal is recorded twice during each experiment. As expected, the AM EPRoC signal exhibits an asymmetric lineshape due to the mixture of absorption and dispersion (see Eq. (1)) that is dependent on the direction of the frequency sweep. If the signal was purely absorption, the shape of the two lines would be symmetric; if it was purely dispersion, they would be "mirrored" since the resonance is passed once from low-frequency to high-frequency and again in the opposite direction. To recover the EPR spectrum, the transient RS-EPRoC signal

---

[2] i.e in excess of the MW carrier frequency $\omega_{mw}$

is *Fourier* deconvolved from the sinusoidally oscillating MW excitation, as explained in detail in the Appendix A. Only the imaginary component of the deconvolved RS-EPRoC spectrum, which corresponds to the imaginary component of the magnetic susceptibility, is shown in Fig. 2b. The CW spectrum of the same sample recorded using a bias current of 5 mA ($B_1 = 27$ µT) is also shown. The different bias currents in the two experiments were chosen to ensure operation in the linear regime, i.e., without microwave saturation.

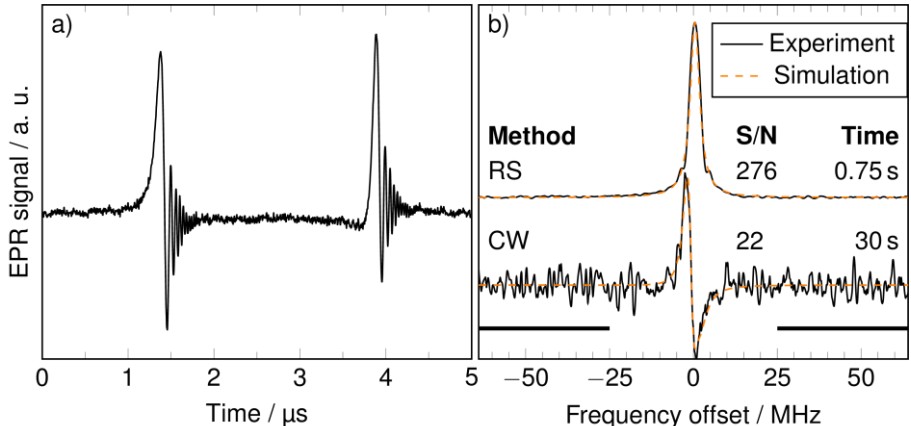

**Figure 2: a) The background-corrected AM RS-EPRoC time trace recorded at a scan rate of 80 THz s$^{-1}$ (corresponding to 2.9 kT/s; $\Delta f_\mathrm{m} = 64$ MHz, $f_\mathrm{m} = 200$ kHz, $I_\mathrm{bias} = 7$ mA ($B_1 = 46$ µT)). b) Experimental data (black) and simulations (orange) of the CW ($I_\mathrm{bias} = 5$ mA ($B_1 = 27$ µT)) and the deconvolved RS spectra.**

As expected from Eq. (1), the CW-EPRoC signal exhibits an asymmetric line shape. There is no asymmetry in the RS-EPRoC spectrum because the complex RS-EPRoC spectrum can be phase-adjusted such that only the absorption signal is visible. In CW-EPRoC measurements, quadrature detection is not possible, and *Kramers-Kronig* manipulation is ill-suited due to slight signal saturation. Both spectra in Fig. 2b were simulated using the "pepper" function of the *Easyspin* software package (Stoll and Schweiger, 2006) assuming a spin-1/2 system with *Lorentzian* broadening. The asymmetry of the line shape in the CW spectrum is included in the simulation via a tailored fitting function according to Eq. (1), using a mixture of absorption and dispersion. A detailed description of the simulations is given in Appendix E. The fit parameters of the CW and deconvolved RS spectrum as well as the measurement parameters for the CW spectrum are given in Appendix D.

The SNR and relevant parameters of CW- and RS-EPRoC measurements are summarized in Table 1. While only the imaginary component of the deconvolved spectrum is shown in Fig. 2, the SNR can in principle be further increased by a factor of $\sqrt{2}$ by the addition of the real and imaginary components of the RS-EPRoC spectrum (Tseitlin et al., 2010). Because the *Kramers-Kronig* relation is needed to obtain the complex transient RS-EPRoC signal in the presented setup, the SNR cannot be increased in the presented setup by the addition of the two spectra. The use of quadrature detection eliminates noise correlation and allows the real and imaginary components to be combined, increasing SNR similar to increasing the number of averages in the collected spectrum. RS-EPRoC measurements yield improved SNR per unit measurement time, and an overall improvement in SNR of nearly two orders of magnitude is obtained. These results are in good agreement with those reported for field-swept RS-EPR of various sample classes, including nitroxyl radicals (Mitchell et al., 2012), irradiated fused quartz (Mitchell et al., 2011a), and samples with long relaxation rates such as a-Si:H or N@C$_{60}$ (Mitchell et al., 2013b; Möser et al., 2017). When comparing the sensitivities in the CW mode between the FM mode of detection and the AM mode of detection, it was found that a discrepancy of about four orders of magnitude between the FM mode and the AM mode is observed. More specifically, the presented EPRoC detector has an FM sensitivity of around $5 \times 10^9 \frac{\text{spins}}{\text{G}\sqrt{\text{Hz}}}$, whereas, in the AM mode the measured CW sensitivity is around $3 \times 10^{13} \frac{\text{spins}}{\text{G}\sqrt{\text{Hz}}}$. This discrepancy in sensitivity partially arises due to the injection-locking of the VCOs, which improves the FM noise floor but not the AM noise floor, accounting for a factor of $\sqrt{12} \approx 3.5$. Only very recently, the noise figure of the implicit AM demodulator was simulated numerically (Chu et al., 2021), revealing a sensitivity of around 1/6 and a degradation in the noise floor of around 20 dB in the frequency range of interest of the implicit AM demodulator,

corresponding to an effective noise figure around 35 dB, i.e., a degradation of around 60 in the spin sensitivity at the output of the AM demodulator compared to the intrinsic SNR of the AM signal with respect the amplitude of the VCO. Together with the factor of 3.5 from above, these factors explain a ~210-fold degradation compared to the FM sensitivity, explaining a large fraction (up to approximately a factor of ten) of the discrepancy between the FM and the AM CW sensitivities of the presented system. As suggested in (Matheoud et al., 2018), an off-chip AM demodulator with a better noise figure may be used to improve the sensitivity of the AM mode detection, preserving more closely the intrinsically identical sensitivities of the FM and the AM modes of detection.

Table 1: SNR for CW-EPRoC and RS-EPRoC methods.

| Method | Bias current, mA | $B_1$, μT | No. of averages | Modulation rate, THz s$^{-1}$ | SNR | Measurement time, s | Normalized SNR, s$^{-1}$ |
|---|---|---|---|---|---|---|---|
| CW-EPRoC | 5 | 27.0 | 1 | 0.5 | 22 | 30.0 | 4.0 |
| RS-EPRoC | 7 | 45.5 | $1.5 \times 10^5$ | 80.4 | 276 | 0.75 | 318.6 |

**3.2 Analysis of the transient RS-EPRoC signal**

RS-EPRoC time traces recorded using four different bias currents (5 mA, 9 mA, 14 mA, 18 mA) corresponding to $B_1$ values of 27 μT, 62 μT, 95 μT, and 118 μT are shown in Fig. 3. The RS-EPRoC time traces were simulated and fit using a solution of *Bloch*'s equations in the steady-state for sinusoidal modulation. For the simulation, *Biot-Savart*'s law and a square-root coil current model were used to calculate the $B_1$ magnitude, which cannot be analytically calculated from the bias current driving the EPRoC sensor (See Appendix E for more information). The simulations were performed using the transient AM RS-EPRoC signals without deconvolution, and the asymmetry of the AM signals was considered by including the quality factor from Eq. (1) in the simulations. The relaxation times of BDPA, $T_1 = 110$ ns and $T_2 = 100$ ns, were taken from the literature (Goldsborough et al., 1960; Mitchell et al., 2011b) and are required for the RS simulations. A thorough description of the simulations is given in the Appendix E.

In Fig. 4, the signal intensities of CW- and transient RS-EPRoC measurements are compared as a function of $B_1$, demonstrating the saturation behavior of the BDPA-benzene complex observed via CW- and RS-EPRoC with rates of $\alpha = 80.4$ THz s$^{-1}$, 201.1 THz s$^{-1}$ and 402.1 THz s$^{-1}$. The CW- and RS-EPRoC signal increases with increasing $B_1$, as expected, and saturation is observed at higher values of $B_1$ for RS- compared to CW-EPRoC experiments. Increasing $\alpha$ leads to a linear regime that extends over several tens of μT, thus allowing the use of $B_1$ values beyond the relaxation-determined limit. Though BDPA is considered rapidly relaxing (~ns), this sample was chosen to facilitate operation in the linear regime for both CW- and RS-EPRoC experiments (see Eq. (4) and (5)). The minimum $B_1$ of the EPRoC is large enough to saturate many slowly relaxing radicals, distorting the lineshape and thereby limiting quantitative analysis. Such samples with slow relaxation, such as single substitutional nitrogen centres ($N_s^0$) in diamonds, a-Si:H or $N@C_{60}$ (Mitchell et al., 2013b; Möser et al., 2017), especially benefit from the RS technique due to the signal saturation that is observed at low MW powers when using CW methods.

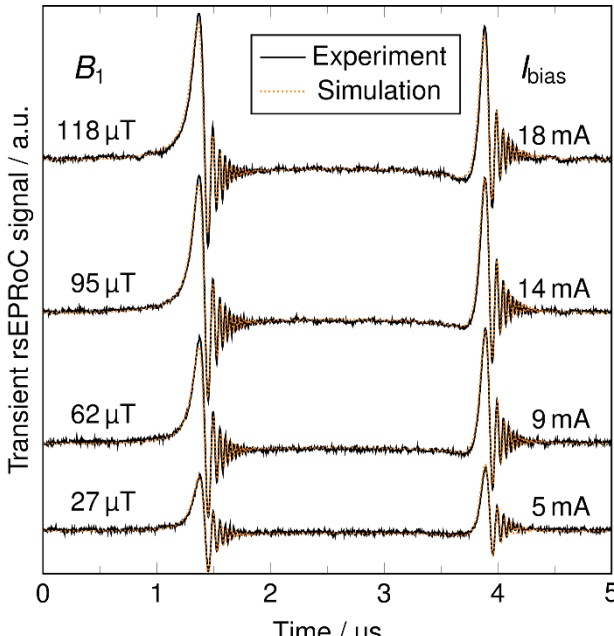

**Figure 3: RS-EPRoC time traces (black) recorded using four different bias currents that correspond to four different $B_1$ magnitudes at a scan rate of 80 THz s$^{-1}$. The spin system passes through resonance twice during each period of the modulation of the MW frequency, see Eq. (6) and (7). The simulations (orange) of the transient acquired data are in good agreement with the experiment.**

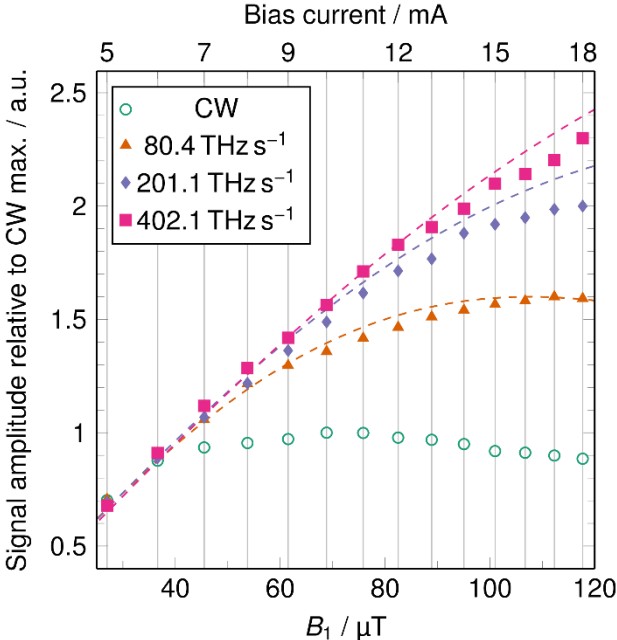

**Figure 4: Signal amplitudes of CW-EPRoC (○) and transient RS-EPRoC for three scan rates (80.4 THz s$^{-1}$, ▲, 201.1 THz s$^{-1}$, ◆, and 402.1 THz s$^{-1}$, ■) as a function of bias current (x-axis, top) and corresponding $B_1$ magnitudes (x-axis, bottom). The dashed lines are simulations of the RS signals.**

Finally, it is necessary to explore the theoretical limits of the RS-EPRoC technique. Figure 5 shows the simulated signal amplitudes of the deconvolved RS-EPRoC spectra as a function of both $B_1$ and scan rate, $\alpha$. The scan rate was increased by increasing scan width while maintaining a constant scan frequency (200 kHz) to ensure that all oscillations have decayed within a single scan period (half-cycle) when considering $T_1$ and $T_2^*$ on the order of 100 ns. The signal amplitudes were normalized to the global maximum of all signals resulting from the simulations to probe the limits of the RS-EPRoC technique with respect to SNR. This analysis extends the rapid scan technique far beyond what is possible with field-swept RS-EPR to encompass a regime that is only accessible via frequency-swept RS-EPR, which has now been implemented with RS-EPRoC. From this simulation, an improvement of the signal amplitude by a factor of about 5 may be achieved compared to the rapid scan measurements presented in this work.

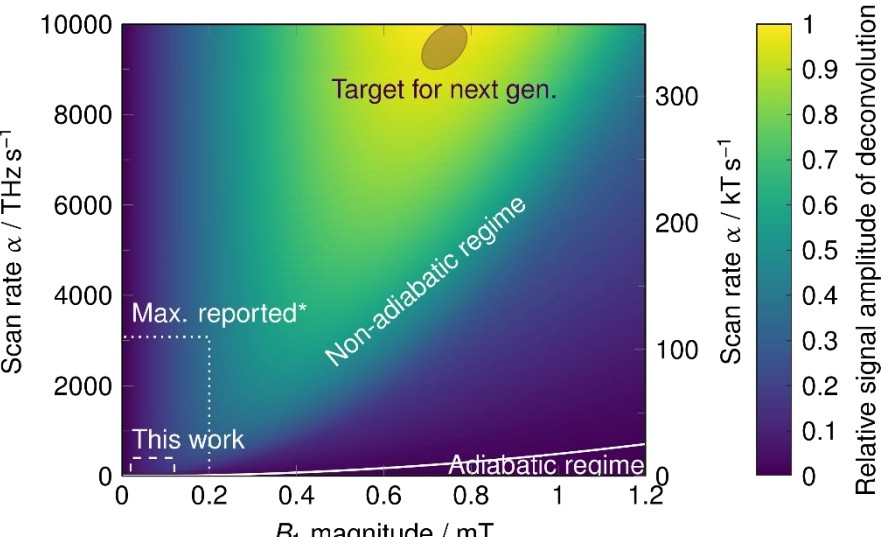

Figure 5: Relative simulated signal amplitude of the deconvolved RS spectrum as a function of both $B_1$ and scan rate $\alpha$. The solid line defines the adiabatic and non-adiabatic regions (Eq. (4) and (5)). The relaxation times were set to $T_1 = 110$ ns and $T_2^* = 100$ ns. The simulation was performed with a constant RS frequency (200 kHz) and increasing scan width (Eq. (7)). The two outlined rectangular regions (dashes) represent the accessible area for the current work as well as that of a study using field-swept RS-EPR where the maximum scan rate[*,3] was reported (Mitchell et al., 2011b). The ellipse shows the target region for the next-gen. EPRoC where the maximum signal is obtained. An improvement of the signal amplitude by a factor of about 5 is expected.

From the simulations, it was determined that simultaneously increasing both $B_1$ and $\alpha$ yields an increase in relative signal amplitude (yellow region in Fig. 5). For a constant $B_1$, an optimal scan rate, $\alpha$, may be achieved that maximizes relative signal intensity without saturation; however, increasing the scan rate when the signal is unsaturated does not increase the signal intensity unless $B_1$ is similarly increased. Likewise, for a constant scan rate, $\alpha$, an optimal $B_1$ may similarly be achieved that maximizes relative signal intensity without saturation, but additional increases in $B_1$ strength without an accompanying increase in scan rate lead to saturation and a decrease in signal intensity due to line broadening. Thus, only an increase of both $B_1$ and scan rate will increase the relative signal amplitude in RS experiments, and this principle will guide further development of RS-EPRoC designs.

In these experiments, the available $B_1$ as indicated by the dashed rectangle in Fig. 5 is limited due to heating of the passively cooled EPRoC detector. If the EPRoC sensor was actively cooled, a $B_1$ of up to 250 µT (~factor of two) is possible with this generation of the EPRoC. In future EPRoC generations with a smaller coil diameter (~100 µm, factor of 2), the $B_1$ magnitude may be increased by an additional factor of two. With the usage of other fabrication techniques than complimentary metal-oxide semiconductor (CMOS), such as bipolar CMOS (BiCMOS) and Indium gallium arsenide (InGaAs), the total $B_1$ gain can be increased by an additional factor of 10 compared to the current generation, resulting in absolute $B_1$ magnitudes of about 1 mT.

The scan rate may be increased by either extending the scan width, which decreases the time spent on resonance, or by using faster repetition rates, which increases the number of full frequency sweeps per unit time. The number of sweeps per unit time; however, is limited by the effective transverse relaxation time $T_2^*$ (Tseytlin, 2017) given by the expression,

$$\frac{1}{f_m} > N \cdot T_2^* \tag{8}$$

with $N$ being in the range of 3 to 5, depending on the amount of acceptable line broadening introduced by Fourier deconvolution. This limit imposes the requirement that the RS signal oscillations or "wiggles" must have decayed completely before the next scan cycle is recorded (Fig. 2a).

---

[3] The fastest scan rate currently reported for a frequency-swept high field/high frequency RS-EPR experiment was 267000 THz s[-1] (Laguta et al., 2018) and is far beyond the limits of this plot.

Currently, the scan rate is not limited by the EPRoC array and its PLL, but by the signal generator supplying the PLL reference frequency (see Sec. 2.2 for a detailed explanation). Commercially available analog signal generators, such as the *Rohde & Schwarz* SMB100B, may improve the scan rate ($\Delta f_{m,max} = 160$ MHz (sweep width 320 MHz or 11.4 mT), (2.5x) and $f_{m,max} = 10$ MHz (10x)). However, as described by Eq. (8) the transverse relaxation time limits the usage of such high modulation frequencies. Additionally, the bandwidth of the PLL limits the modulation frequency to about 5 MHz, such that an improvement of the scan rate of a factor of 5 is realistic. The next-generation EPRoC with on-chip PLLs and higher bandwidths of up to 80 MHz is currently in development and will be capable of delivering scan rates of up to $10^4$ THz s$^{-1}$ via scan widths of more than 2.4 GHz (85.6 mT) and repetition rates of 2 MHz or more. Due to the larger bandwidth of the PLL and a different PLL design where the FM signal may be extracted without filtering, the FM signal may additionally be used for data analysis exploiting the advantage of the array giving access to a larger sample volume and hence increased concentration sensitivity.

## 4 Conclusions

In this work, the use of VCO-based EPRoC detectors is introduced for closed-loop non-adiabatic RS-EPR experiments. By embedding the VCO into a large bandwidth PLL and using the implicit amplitude demodulation capability of current-biased LC tank oscillators, the experimental setup of RS-EPRoC experiments is comparatively simpler compared to conventional field-swept RS-EPR. In these experiments, an improvement in SNR of almost two orders of magnitude is achieved compared to CW experiments performed using the same EPRoC detector. The improvement in SNR arises from a combination of an increased signal amplitude due to a later onset of sample saturation (a factor of approximately 2x) in the RS regime and an improved noise floor due to the significant signal averaging employed in the RS measurements. With these experimental results, it is confirmed that – similar to field-swept RS-EPR – in RS-EPRoC the RS signal is less prone to $B_1$ field saturation and remains in the linear $B_1$ regime up to 90 µT for BDPA at the fastest scan rate investigated (402.1 THz s$^{-1}$). The time-domain signals can be reliably transformed to depict the EPR susceptibility. Although the reported CW sensitivities are greatly inferior to the FM sensitivities of the presented chip, most of this discrepancy can be explained by the poor noise figure of the implicit AM demodulator. Therefore, by using improved, low-noise AM demodulators in the future, it is expected that AM sensitivities similar to those observed in the FM mode may be obtained, allowing the full benefits from the simplified experimental setup and the large SNR gain in the AM rapid scan mode of detection to be realized.

The inherently large frequency sweep width capability of the EPRoC array with sweep widths of up to 2.4 GHz (86 mT) and intrinsically near-constant detection sensitivity will allow investigations of transition metal ions and other broad line spectra by RS-EPR. The ability to use small permanent magnets via frequency swept RS-EPR, coupled with its small size and power consumption, makes EPRoC applications very flexible. In the future, EPRoC detectors may be integrated into various complex and harsh sample environments enabling *in situ* and *operando* EPR measurements that have previously been inaccessible. This includes hand-held devices for in-the-field multiline fingerprinting applications in chemistry, medicine, biology, material science, and physics.

## Appendices

## Appendix A: Fourier deconvolution

The *Fourier* deconvolution procedure was published in detail in references (Stoner et al., 2004; Joshi et al., 2005b; Tseitlin et al., 2011a; Tseytlin, 2017) and is briefly summarized here. To obtain the EPR spectrum, the RS signals must be *Fourier* deconvolved from the frequency spectrum of the MW excitation. Assuming a linear response $r(t)$ of the spin system under the influence of the excitation $d(t)$ and $B_1$ small enough to avoid saturation, the following is obtained:

$$r(t) = (h * d)(t) = \int_{-\infty}^{\infty} h(\tau)d(t - \tau)\mathrm{d}\tau \tag{9}$$

where $h(t)$ is the impulse response of the spin system (often referred to as "wiggles"), and $*$ denotes the convolution operator. The driving function (Tseitlin et al., 2011a) for the RS modulation, $d(t)$, is then defined as:

$$d(t) = \exp\left(i\int_0^t \omega(\tau)\mathrm{d}\tau\right) \tag{10}$$

where $\omega(\tau)$ is the time-dependent angular MW frequency (i.e. the waveform) and its integral is the MW phase. In the frequency domain, the convolution in Eq. (9) becomes a multiplication:

$$R(\omega) = H(\omega)D(\omega) \tag{11}$$

where $R(\omega)$, $H(\omega)$, and $D(\omega)$ are the *Fourier* transforms of $r(t)$, $h(t)$, and $d(t)$, respectively. Thus, the EPR spectrum can be obtained in the frequency domain by a division as:

$$H(\omega) = R(\omega)/D(\omega) \tag{12}$$

The algorithm of the deconvolution procedure is as follows: The zero-padded transient baseline-corrected RS signal is *Fourier* transformed with a *Welch* apodization window (Welch, 1967). The excitation function is calculated assuming a sinusoidal frequency scan, numerically integrated, zero-padded and *Fourier* transformed with the same *Welch* apodization window. According to Eq. (12), the EPR spectrum containing both real and imaginary components of the complex susceptibility is obtained by the division of both *Fourier* transforms. The zero-padding function improves frequency resolution while the apodization window avoids sharp transitions to zero when zero-padding which would result in spikes in the *Fourier* transforms.

**Appendix B: Bandwidth of the transient RS-EPRoC signal and its relation to the PLL bandwidth**

The bandwidth of a transient RS-EPR signal for a single *Lorentzian* may be calculated from the scan rate $\alpha$ in Hz/s and the effective transverse relaxation time $T_2^*$ (Mitchell et al., 2012),

$$\mathrm{BW}_{\mathrm{signal}} \approx N\alpha T_2^* = 2\pi f_{\mathrm{m}} \Delta f_{\mathrm{m}} T_2^* \tag{13}$$

where N is a parameter that describes the acceptable line shape broadening and is usually between 3 and 5. The signal bandwidth of the transient RS-EPR signal is determined by the spacing of the "wiggles", which are a measure of the resonance offset, the modulation frequency, $f_{\mathrm{m}}$, and the modulation amplitude, $\Delta f_{\mathrm{m}}$. The spacing of the "wiggles" on the trailing edge of the transient RS-EPR signal at constant $T_2^*$ and constant $f_{\mathrm{m}}$ gets smaller with increasing modulation amplitude, $\Delta f_{\mathrm{m}}$, since the resonance offset gets larger. The linear dependence of the signal bandwidth on $T_2^*$ can be explained by the fact that the "wiggles" are visible for a longer time. Since the resonance is passed twice in one full RS cycle, only half of the available bandwidth of any detection system is available for the signal present in each half cycle, such that the BW of the detection system $\mathrm{BW}_{\mathrm{detection}}$ should be twice as large as the signal bandwidth as

$$\mathrm{BW}_{\mathrm{detection}} \geq 2N\alpha T_2^* \tag{14}$$

In ref. Mitchell et al. (2012), relation (14) was used to determine the quality factor needed for detection of an undistorted RS-EPR signal. Concerning the EPRoC, the bandwidth of the PLL, about 10 MHz, limits the bandwidth of the FM signal to about 5 MHz. Using a conservative estimate for $N = 5$, and a $T_2^*$ of 110 ns, the signal bandwidth needed for an undistorted FM signal is about 80 MHz. Since the available bandwidth is much less than the required signal bandwidth, the FM signal was not considered in these experiments.

**Appendix C: Digital post-processing of the EPRoC spectra**

Both, CW- and RS-EPRoC spectra are digitally filtered with a moving average, 2nd order *Savitzky-Golay* filter. The filter window is adjusted such that the linewidth is broadened by less than 5%. For CW data, the effective acquisition time is calculated from the number of data points of the sweep, $N_{\text{points}}$, and the time constant of the lock-in amplifier, $\tau_{\text{LIA}}$, as

$$T_{\text{acq,cw}} = 3\,N_{\text{points}} \cdot \tau_{\text{LIA}} \tag{15}$$

A factor of 3 is introduced to take into account the re-arm time of the lock-in amplifier required to achieve 99.9 % of the maximum signal intensity. For RS experiments, the effective acquisition time is calculated using the number of averages, $N_{\text{avg}}$, and both the number, $N_{\text{fc}}$, and the period, $T_{\text{fc}}$, of all RS cycles present in the signal acquisition, respectively, as

$$T_{\text{acq,rs}} = N_{\text{avg}} N_{\text{fc}} T_{\text{fc}} \tag{16}$$

The signal amplitude of the CW measurements is defined as the peak-to-peak amplitude of the AM signal. The root-mean-square (RMS) noise is determined from the baseline regions of the spectrum (see Fig. 2b). For both CW- and RS-EPRoC measurements, ~61 % of the data points were used for calculation of the RMS noise. The SNR is calculated as the ratio of the signal amplitude to the RMS noise. The signal amplitude of the deconvolved RS-EPRoC spectrum is defined as the maximum value of the imaginary part of the deconvolved RS spectrum. The RMS noise is calculated for the CW measurements from the baseline regions of the spectrum. The SNR is calculated as the ratio of the signal amplitude to the RMS noise. For the saturation analysis, the signal amplitude of the RS measurements is defined as the peak-to-peak amplitude of the transient RS signals since a deconvolution of the highest scan rate was not possible due to overlapping signals.

To compare the signal amplitudes of different methods and scan rates, the relative signal amplitude is used.

**Appendix D: Fit and CW measurement parameters for Figure 2**

The *Lorentzian* peak-to-peak linewidth of the fit of the deconvolved RS-EPRoC spectrum is 1.98 MHz (0.071 mT). The fit parameters of the CW-EPRoC spectrum are: *Lorentzian* peak-to-peak linewidth: 2.42 MHz (0.086 mT), $Q_{\text{coil}} = 3.54$ as defined in Eq. (1).

The measurement parameters of the CW spectrum are $f_{\text{m}} = 100$ kHz; $\Delta f_{\text{m,pp}} = 0.768$ MHz (0.028 mT), lock-in time constant, 10 ms and filter order, 24 dB, which gives an effective noise bandwidth of the LIA of 102.62 Hz.

**Appendix E: Simulation of transient RS-EPR signals**

All simulations of transient RS-EPR signals were performed by numerically solving the *Bloch* equations (Tseitlin et al., 2013; Stoll and Schweiger, 2006) in the steady state using *Easyspin*'s *blochsteady* function. A spin-1/2 system with a g-value of 2.003, a *Lorentzian* line shape, and relaxation times $T_1 = 110$ ns and $T_2 = 100$ ns were assumed based on previous reports for BDPA (Goldsborough et al., 1960; Mitchell et al., 2011b).

**1 Simulation of the transient RS-EPRoC signals to obtain B₁ magnitude**

Since the range of the $B_1$ magnitude of the EPRoC is not precisely known and cannot be measured by *Rabi* oscillations due to the limited bandwidth of the PLL, 14 transient AM RS signals were recorded with increasing bias current and simulated as described in in the preceding section without *Kramers-Kronig* manipulation and without subsequent deconvolution. Thus, the lineshape asymmetry expected from Eq. (1) was also considered by using a tailored fitting function in the simulation according to Eq. (1) that includes contributions from both absorption and dispersion. To convert the bias current to a $B_1$ magnitude, a two-parameter square-root model of the current in the coil, $I_{\text{coil}}$, taking the curvature of the coil current at low bias current into consideration, was assumed as

$$I_{\text{coil}} = a + b\sqrt{I_{bias}} \tag{17}$$

From this, the $B_1$ magnitude, as seen in Fig. E.1, was calculated assuming a circular single turn inductor with a radius, $R$, of 100 µm using *Biot-Savart*'s law as

$$B_1 = \frac{1}{2}\,\mu_0\,\frac{I_{\text{coil}}}{2R} = \frac{1}{2}\,\mu_0\,\frac{a+b\sqrt{I_{bias}}}{2R} \tag{18}$$

where $\mu_0$ is the vacuum permeability. The factor of ½ takes into account that only half of the $B_1$ field is available for microwave excitation due to the two counter-rotating microwave fields in the rotating frame.

For the simulation of the AM RS-EPRoC signals, three global parameters were slightly varied for all transient RS signals - the parameters $a$ and $b$ and the quality factor $Q$ of the VCO from Eq. (1).

## 2 Simulation of the RS-EPR signal amplitude as a function of B₁ and scan rate

For each point in Fig. 5, a complex transient RS-EPR signal containing both dispersion and absorption was simulated as described in the preceding section and subsequently deconvolved. From each deconvolution, the signal amplitude, which is the maximum of the absorption signal, was extracted. The obtained values were normalized to the global maximum of all the signal amplitudes to form a relative comparison.

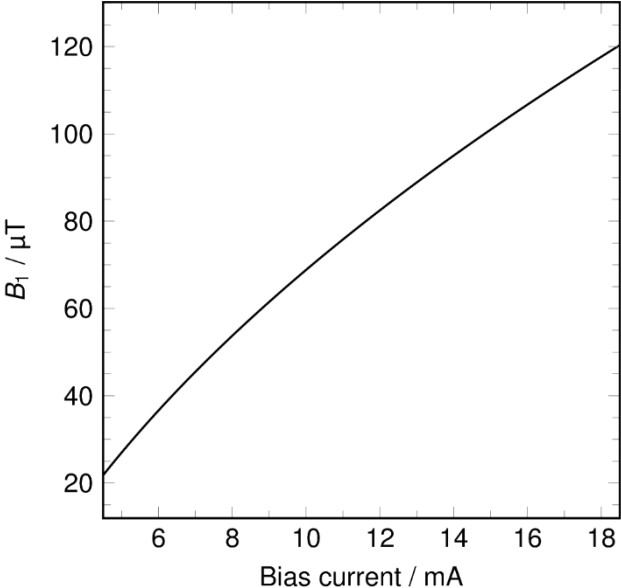

**Figure E1: B₁ magnitude obtained by the square-root model of the bias current used throughout the EPRoC experiments reported in this manuscript.**

## Appendix F: Determination of sample volume and mass

The sample volume was approximated using multiple photographs of the sample as shown in Fig. 1b while varying the light present to differentiate shadows from the sample material. To calculate the sample volume, a cuboid was assumed. The planar

dimensions of the cuboid were determined from the shape of the sample in the photograph while height was determined using its shadow on the chip. In this way, the sample volume might be overestimated. The density of the BDPA-benzene complex is 1.220 g/cm³ (Azuma et al., 1994). From the volume and the density, the sample mass and the number of spins were calculated.

## Simulation and data processing availability

All programming used for data analysis, simulation, and processing is available from the authors upon request.

**Data availability**

The data that support the findings of this study are available from the corresponding authors upon request.

**Author contribution**

SK, BN, AS, JA and KL defined the goals of the research and designed the experiments. SK performed all EPR experiments, data processing, and simulations. AC and JA designed the EPR-on-a-Chip spectrometer. SK, JM, BN and JA evaluated the results of the experiments and wrote the manuscript. KD advised the authors at all stages of the research and authoring of the manuscript. The paper was revised by all authors.

**Competing interests**

The authors declare that they have no conflict of interest.

**Acknowledgements**

We are grateful to Jannik Möser and Jason Sidabras for experimental support and for helpful discussions. This work has been supported by the Bundesministerium für Bildung und Forschung under contract number 01186916/1 (EPRoC) and by the HEMF (Helmholtz Energy Materials Foundry) infrastructure funded by the Helmholtz association (HGF). B.N. acknowledges the financial support from the Deutsche Forschungsgemeinschaft (project numbers 410866378 and 410866565).

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
