# Peer review of "Rapid Scan Electron Paramagnetic Resonance using an EPR-on-a-Chip Sensor"

_Magnetic Resonance, 2021_

## Author Comment (AC1)

**Rapid Scan Electron Paramagnetic Resonance using an EPR-on-a-Chip Sensor**

**Reply from the authors**

Silvio Künstner and co-authors

Dear colleagues,

My co-authors and I would like to thank Daniella Goldfarb as editor and the reviewers for their efforts in evaluating our manuscript. We are pleased that the reviewers' comments are generally positive and provide valuable suggestions to improve the manuscript even further. In response, we revised the manuscript as detailed below (reviewers suggestions/remarks are indented, our replies not indented). Please note that the numbers of equations and line numbers changed with respect to the preprint.

**RC1**

**General comment**

> The manuscript presents a new method that combines the use of EPR on a chip technology with rapid scan (RS) approach in EPR. EPR on a chip uses a small microwave oscillator which is based on active microwave circuit coupled to LC circuit with inductive loop on which the sample is placed. EPR signal is recorded as changes in the oscillator frequency and/or amplitude at the resonance condition. RS with EPR on a chip can be very advantageous since instead of scanning the magnetic field, which has many limitations, one can scan the frequency without the need to have a low Q resonator. In general, the paper is well-written and presents nice experimental results. I have one major comment and few minor comments as follows:

We thank the reviewer for the kind words about our work and the manuscript.

**Major comment**

> The paper makes some claims about spin sensitivity, which are not convincing. It uses a test sample of BDPA with about 2x10^15 spins (this number is not written in the paper, but can be calculated using the data given), and shows measurements with SNR of 236 and then claims, based on data from another paper, that the absolute spin sensitivity of the setup is 6x10^7 spins. Same problem with the claims for concentration sensitivity. I am afraid this looks very unconvincing. The authors should either present clear experimental evidence for their claim spin and concentration sensitivities, or tone down their claims.

The calculation of the RS spin sensitivity was removed from the manuscript. The sensitivity for the AM signal of the CW measurement calculated from the SNR of the spectrum, the number of spins in the sample and the effective noise bandwidth of the detection system is now stated, which is of the order 10^13 spins/G/sqrt(Hz). In addition, the FM sensitivity is stated, which is of the order of 10^9 spins/G/sqrt(Hz). In the revised version of the manuscript, we now provide some reasoning why the AM sensitivity is worse than the FM sensitivity and also include more information about the AM detection using VCO-based EPRoC detector. Also, the number of spins in the sample (~10^15 spins) is now part of the sample description.

**Minor comments:**

Line 33: Conventional EPR employs two types of experimental procedures. High Q is good mainly for CW.

We added the information that CW benefits from high Q.

Line 43: kEuro and not TEuro.

The term was corrected.

Line 55: Suggest to also cite related works, such as : "A Single-Chip Electron Paramagnetic Resonance Transceiver"" and "An Ultrasensitive 14-GHz 1.12-mW EPR Spectrometer in 28-nm CMOS"

We added the proposed citations and added some information on the different detection principles.

Line 105: Less than 10 ppm is not that simple.. and also temperature stability is not simple..

We removed the statement of the homogeneity of the magnet to make the statement more general.

Line 110: what is the max frequency of the AM demodulation?

The bandwidth of the implicit AM demodulator is a few hundred MHz (roughly 600 MHz). The information was added to the manuscript. Also, a more detailed description of the AM demodulation was added to the text in Sec. 2.1.

Line 114 and other places: The claim for compactness of the system and the use of Rohde & Schwarz SMB100A and Anfatec eLockIn 203 and Zurich Instruments UHF-LIA as part of the setup seem to be conflicting.

We added the term "proof-of-principle" to the last sentence of the introduction to explain that we do not (yet) have a completely miniaturized spectrometer (yet).

Line 115: what is the minimal B1 that can be used to sustain working conditions for the VCO?

We added the minimal B1 of about 27 µT in the corresponding sentence.

Line 121: What is the number of spins n the sample?

The number of spins (~10^15) is now stated in the text.

Line 122: When referring to Appendices, please mention which Appendix.

All appendices are now specifically referenced in the text.

Line 130: This discussion should come before mentioning AM modulation above.

The corresponding section was moved and extended to also include the remarks from RC2.

Line 134: Possible cite this ref from Arxiv?

The article is now available as preprint of Magnetic resonance and is correctly cited.

Line 140: Try to be more quantitative, what bandwidth you have, what is needed, etc..

The numbers are now given in the text with a reference to Appendix B, where the calculation is explained.

Eqs 3 and 4: not clear why the authors talk about two types of conditions.

The two conditions are now mentioned in the discussion of Fig. 4.

Line 165: Missing "of a"

The words "of a" were added to the corresponding sentence.

Fig. 4: Is this plot for the same total acquisition time? bandwidth of detection? Is the amplitude and SNR are comparable?

The total acquisition time was different for the three saturation curves. The number of averages, however, was 50'000 for all curves. The sampling rate was 450 MHz for all time traces.

Line 250: delete "is"

The word "is" was removed.

Eq (9) : Please briefly explain why the driving function need to have "memory" to previous time periods and not simply reflect the frequency of excitation at a given time

We added an explanatory sentence to the Appendix A.

Eq (12): This equation does not look intuitive. If T2 is very large the signal is changing slowly as you scan the frequency. Possibly it can be explained in 1-2 sentences.

This equation is now explained in the Appendix B.

Line 338: re-arm?

We have removed the reference to the re-arm time in the main text and have given a description of the re-arm time of the lock-in amplifier in Appendix C.

**RC2**

**General comments**

The manuscript present, for the first time, rapid scan measurements performed using a single-chip integrated oscillator. This approach was proposed and very briefly discussed in Ref. (Gualco et al., 2014), but not yet demonstrated experimentally. Contrary to the majority of previously reported works on the rapid scan, the rapid scan in this work is implemented as rapid frequency scan instead of rapid field scan. This is technically possible and very efficiently implemented because a microwave oscillator is used instead of a microwave resonator combined with a microwave source as in conventional EPR spectrometers. In the current implementation a scan range of 64 MHz at the maximum frequency of 1 MHz, which corresponds to a scan rate of 400 THz/s, has been demonstrated. This scan rate is slower that the best results reported to date for the rapid field scan. However, as claimed also by the authors, I believe that significant improvements are realistic. The single chip frequency rapid scan is, indeed, well suited to achieves scan widths, scan speeds, and scan rates well beyond the current limits of the magnetic field rapid scans. The EPR signal is detected as a variation of the oscillation amplitude as a function of the oscillation frequency. In principle, the measurement of the variation of the oscillation frequency would also be possible but, I guess, practically more complicated because the frequency variation due the EPR resonance would be much smaller than the frequency scan width, creating significant problem of "dynamic range" (which are difficult, although not conceptually impossible, to overcome). It is also important to underline that one of the major problems present in several of the previously reported single-chip integrated oscillator EPR detectors is the relatively large minimum B1, which creates saturation problems in conventional CW slow scan experiments. The use of the rapid scan overcome, at least partially, this issue since the optimum conditions are achieved with a larger B1. The rapid scan approach demonstrated here is certainly a very important milestone in the application of single-chip integrated oscillator as EPR detectors. For this reason, the manuscript certainly deserve to be published.

We thank the reviewer for their detailed comments, which greatly helped us to improve our manuscript.

**Major specific comments**

Abstract, Figure 2, lines 190-195, conclusions: The way the spin sensitivity is computed is not clear to me. The authors use a BDPA sample of 0.67 nL. From the spin density of BDPA (about 1.5x10^27 spins/m3), the number of spins is about 10^15. Since the measured SNR is about 236 in a measuring time of 0.75 s (Figure 2), the spins sensitivity seems to me something like 4x10^12 spins/sqrt(Hz),

whereas the one declared in the paper is 6x10^7 spins/sqrt(Hz). Since the difference is more than 4 orders of magnitude, I think there is something not correct or unclear in the author's reasoning. The reasoning of considering the previous results obtained with the frequency variation and extrapolate it to this case of amplitude detection based on the ratio in SNR between the CW and RS experiments performed here seems to me not "conceptually" correct (and not compatible with the results shown in Figure 2).

Lines 203-210: Also this part of the "sensitivity discussion" is not clear to me. In particular the discussion of the the PSD and RMS noise values are not clear to me. PSD and RMS noise are two different quantities related by an integral once the integration bandwidth is properly defined. So the phrase "...the PSD noise is usually better than the RMS noise of an EPR spectrum..." does not make sense to me. I would suggest to the authors to define clearly how the experiment is performed (including the analog bandwidth and the digital processing) and the way they have computed the spin sensitivity from the processed data. This should be enough to compare it to other papers knowing the different way the experiments are carried out (CW, RS, pulsed), the experimental parameters (analog filtering, digital filtering, etc. etc.), and the given definition of the spin sensitivity. If the experimental conditions and parameters are properly described, each reader can easily "renormalize" them to his/her own sensitivity "definition".

The calculation of the RS spin sensitivity was removed from the manuscript. The sensitivity for the AM signal of the CW measurement calculated from the SNR of the spectrum, the number of spins in the sample and the effective noise bandwidth of the detection system are now stated, the former is of the order 10^13 spins/G/sqrt(Hz). Additionally, the FM sensitivity is now stated, which is about 10^9 spins/G/sqrt(Hz). We also explain why the AM sensitivity is worse than the FM sensitivity. Also, the number of spins in the sample (~10^15) is now part of the sample description.

Lines 111: It is not clear if the voltage variation measured in this work is equal to the oscillation amplitude variation at the resonator ? This is a necessary information to evaluate if the amplitude detection implemented here can or not, in practice (and not in theory where effectively they should be similar in the respective optimized conditions) achieve the same spin sensitivity as the frequency detection reported previously using the same chip. To complete the comparison the frequency and amplitude noise spectral densities should also be also considered. This point is, of course, linked to the previous one. I wonder if the voltage amplitude measurement performed here is not "sub-optimal" (i.e., the voltage variation is significantly smaller than the voltage variation at the resonator, which in turn makes the spin sensitivity worse that in the case of the frequency variation detection if the voltage noise is not reduced by the same factor).

We agree that the given information about the AM detection was not detailed enough in the first version of the manuscript. Therefore, a more thorough description concerning the AM-sensitive detection has been added to the manuscript in Sec. 2.1 of the manuscript.

Figure 2, Figure 3, lines 355-358: Why the two signals in Figures 2 and 3 are not identical ? I guess that it is because there is a mix of absorption and dispersion which gives non- identical signals when the frequency is scanned up or down (pure absorption signals would be have the same shape in the scan up and down, pure dispersion signals would have "mirror shapes" in the scan up and down). Please comment on this in the manuscript and write the details of the simulation in the Appendix E. It seems to me that the reported simulation results are not a result directly taken from EasySpin. Are the EasySpin absorption and dispersion signals combined with an appropriate phase shift maybe computed from an estimation of the Q-factor (as suggested by Equation 1) ? Or maybe a circuit simulator is used where the sample is modeled by a coupled resonator. This would be correct "quantitatively" for a CW slow passage at low B1 but I guess not for a RS.

The reviewer correctly explained the reason for the asymmetry, which was indeed missing in the manuscript. We now provide an explanation in the updated version of the manuscript. Additionally, the description of the simulation in Appendix E was extended to better explain the procedure.

Figure 4 and Figure 5: In terms of precessing magnetization (i.e., Mxy), the maximum value for T1=T2 and the optimum level of B1 and scan speed is: (1/2)*Mo for the CW and Mo for the RS. So, in terms of precessing magnetization, the difference is 2 (and not 5). Of course, depending on the way the CW and RS signals are computed (field modulation amplitude, peak-to-peak or amplitude, etc. etc.) the ratio can be different. But, I would prefer to consider a more "fundamental" quantity which is the precessing magnetization. Of course, in practice the optimal condition for the CW with field modulation is obtained with a B1 and a field modulation amplitude which determines linewidth broadening, which might or not be "tolerable".

We agree with the reviewer that the "fundamental" quantity for the signal intensity is the precessing magnetisation. As already mentioned in the comment, the factor of 2 that we may gain with RS compared to CW is a theoretical concept since it only involves the precessing magnetisation. In this view, the spin system is completely saturated at (1/2)*Mo in CW. At this point, however, we do experience considerable line broadening in the spectrum, which complicates quantitative analyses.

Additionally, we added an explanation of the factor 5 (amplitude gain with higher B1 and faster scan rates), which was misleading in the preprint. The factor 5 may be gained considering the rapid scan measurements only as shown in the manuscript.

**Minor comments**

Line 43: Typo: "50 kEuros" instead of "50 TEuros".

We corrected this mistake.

Line 84: I agree that the approach proposed here could allow in the future to perform rapid scans larger than 20 mT (600 MHz). However in this manuscript it

is demonstrated up to about 64 MHz. I think that this should be mentioned also here.

It is now mentioned in the text that we use a much smaller sweep width in the experiment. For that, we added the term "proof-of-concept" to the manuscript.

Line 99: I think that it should be mentioned that "The rapid scan with single-chip integrated oscillators was proposed and briefly discussed in Ref. (Gualco et al., 2014), but not yet demonstrated experimentally. Here we report ....."

We added a similar statement to the manuscript.

Line 105: "< 10 ppm". Please specify on which volume you are considering <10 ppm homogeneity.

The statement about the homogeneity was removed and replaced by a more general statement.

Lines 159 and 281: The reason why the "..in these experiments was limited by the RF generator to ...." is not very clear. I would suggest to add a couple of sentences to clarify this point. I guess this is related to the chip architecture and, in particular, to the way the frequency scan is implemented (more complex, clever, and efficient that a simple voltage externally applied to the integrated varactor).

We added a better explanation to the manuscript to describe the limitation of the RF generator. Additionally, we added more information about the chip architecture in Sec. 2.1.

Line 250: Typo: "...about 5 is may be.." instead of "...about 5 may be.."

We removed the word "is".

Line 120: The minimum value of B1 produced by the chip is 27 uT or so. BDPA has significant saturation from B1 in the order of 100 uT or so. So the choice to use only BDPA as sample for this work does not allow to show one of the advantages of the RS when applied to the single-chip approach. The minimum B1 is often relatively large and might cause significant saturation in the conventional CW slow scan. I would suggest the authors to, at least, comment on this point (even if obvious for an expert). Although definitely not "necessary" and "important" for this manuscript, an RS experiment on a sample which is "deeply saturated" in the conventional CW slow scan mode would be a nice addition to the manuscript. A less elegant but maybe still valid example could be the use a very small sample of BDPA placed in close proximity to the coil wire where B1 is significantly larger to show that the RS scan can solve this saturation issue.

We agree that BDPA is not the ideal signal to demonstrate the benefits of RS due to its fast relaxation times. However, to facilitate an accurate and fair comparison between CW and RS, a sample that does not saturate at the fairly large B1 values is needed. Therefore, we added an explanation in the discussion of Fig. 4 explaining that BDPA is not the optimal sample to show the benefits of RS-EPR due to its fast relaxation rates.

---

## Author Response (AR2)

Dear colleagues,

My co-authors and I thank Daniella Goldfarb as editor and the reviewers for their efforts in re-evaluating our manuscript. We are pleased that the reviewer's comments are generally positive, and we appreciate their useful suggestions for improving the manuscript. In response, we edited the manuscript as detailed below. The reviewer's suggestions/remarks are indented while our replies not indented.

**RC1**

**General comment**

The authors have very significantly improved the manuscript. In particular all my suggestions/remarks (maybe except one, see below) have been addressed properly and clearly.

We thank the reviewer for their kind words about our work and the revised manuscript.

**Major comment**

Table 1: I still do not understand how it is possible to obtain an improvement in SNR of a factor 68 between the CW and RS experiments for a sample with T1=T2=100 ns and B1=25 uT (i.e., in almost optimal conditions for the CW experiment), and with a CW modulation frequency of 100 kHz. Since, in these conditions, the precessing magnetization in RS can be maximum a factor of 2 larger than in CW, such additional factor 34 (or more) should come from the noise.

But the CW experiment is performed with a modulation at 100 kHz, so I guess the noise spectral density in the effective bandwidth of the RS experiment is probably not much smaller. In order to clarify this point, I would suggest to the authors to show the noise spectral density and, eventually, the signal voltages both "scaled" at the detection point (i.e., at the point Vx) for the two experiments.

[Figure]

*Figure 1 Simulated detector noise*

We agree with the reviewer that the SNR improvement can only originate from an improvement in signal amplitude (a factor of 2x due to reduced saturation) and an

improved noise floor. As it turns out, the low-frequency noise behavior of the intrinsic AM demodulator that has been used in this paper is relatively poor, as can be seen from Fig. 1 above, which shows the simulated AM noise of the VCO in its output signal compared to the noise inside the demodulated signal that we use in the paper (AM output for CW, *purple*, and RS, *red,* diff output is the noise floor in the differential oscillator output, AM output is the noise in the demodulated output). In the CW measurement, the lock-in amplifier extracts the noise around the modulation frequency of 100 kHz with the very small equivalent noise bandwidth. By contrast, the noise floor in the transient RS data is the integrated noise from the repetition rate of the RS frequency sweep to the full bandwidth of the detection system (200 kHz to 128 MHz). Due to the decreasing noise vs. frequency characteristic of Fig. 1, the RS noise floor is improved compared to a CW experiment. Here, although the integrated RS noise is much larger than the CW noise in a single scan, in the RS experiments, the noise floor can be reduced by significant averaging (~150,000 averages). For a (somewhat) fair noise comparison, one could average the same amount of time that is needed to acquire a single CW point.

We have calculated the anticipated SNR difference between CW and RS, given the data in the above figure, taking into account the different signal processing schemes for CW and RS mentioned above (narrowband filtering in CW, wide bandwidth detection and averaging in RS). These calculations indicate an improvement of a factor of 3.8x in the noise floor of the RS experiment compared to the CW measurement for the same measurement time. This corresponds to an agreement between the measured (not-normalized) SNR improved of 12.5 and the simulated improvement of 6.3 within a factor of two. At this point, we would like to thank the reviewer for their question, which has brought us to a deeper understanding of the different mechanisms that establish the noise floor in CW vs RS EPR experiments with the oscillator-based EPRoC detectors.

> In the answer, the authors mention: "At this point, however, we do experience considerable line broadening in the spectrum, which complicates quantitative analyses". I don't understand this answer. I would not expect a "considerable" line broadening for B1=25 uT with T1=T2=100 ns. If there is "considerable line broadening", I would suggest to discuss this in the manuscript (is it due to oscillator frequency noise due to the unfiltered VCO control voltage ?).

We apologize for the confusion. The sentence mentioned by the reviewer "At this point, however, we do experience considerable line broadening in the spectrum, which complicates quantitative analyses," refers to our statement: "In this view, the spin system is completely saturated at (1/2)*Mo in CW." It does not refer to the experimental/sample parameters of the measurements shown in the manuscript for which virtually no sample saturation is present.

> Line 343: I do not understand the meaning of "....from an increased signal amplitude due to a later onset of sample saturation in the RS regime". It seems to me that the saturation arrives effectively at a larger B1 in RS but overall the signal amplitude improvement in RS, as discussed also above, should be factor of 2 or so with respect to CW operated at the optimal B1 and for T1=T2.

Thank you for bringing this point to our attention. The sentence "The improvement in SNR arises from an increased signal amplitude due to a later onset of sample saturation in the RS

regime and an improved noise floor at higher frequencies of the EPRoC detector." Was included to explain the SNR improvement as partially coming from an increased signal (factor of 2x) and partially from an improved noise floor which is obtained via signal averaging. We have revised this sentence to read as follows,

"The improvement in SNR arises from a combination of an increased signal amplitude due to a later onset of sample saturation (a factor of approximately 2x) in the RS regime and an improved noise floor due to the significant signal averaging employed in the RS measurements."

to avoid this confusion in the final version of the manuscript.

**Minor comments:**

Line 26: Reading this phrase, it seems that a larger sweep range would affect the "sensitivity". I would suggest the authors to rephrase it to make it more clear. I guess that the "spin sensitivity" will not be affected, but I agree that a larger sweep range would extend the applicability of the proposed method to a larger class of samples.

A larger sweep width will be beneficial if the B1 magnitude can be increased in the same manner, and a sample with longer relaxation times is used, such that an increase in the scan rate and an improvement of the spin sensitivity may be seen. Without the additional increase in B1; however, the reviewer is correct in saying that no increase in spin sensitivity is expected for the same repetition rate. We have elected not to alter the sentence structure in the manuscript because the maximum B1 of the current EPRoC was not utilized due to this sweep width limitation. The reviewer is also correct in stating that a larger sweep width indeed improves the applicability to a broader class of samples in the absence of spin sensitivity improvements.

Line 129: Typo: "and an" instead of "andan"

This typo was corrected.

Line 140: The fact that the amplitude and frequency mode should in principle provides the same sensitivity was discussed in details also in Matheoud et al. (2018) (Matheoud et al., Journal of Magnetic Resonance 294 (2018) 59–70). I would suggest to add this reference here (a reference which is already cited in the manuscript).

The citation has been added to the corresponding sentence.

Line 348: Typo: "by" instead of "bu".

This typo was corrected.